# Diagnostics of preeclampsia based on Congo red binding to urinary components: Rationales and limitations

Sergei A. Fedotov[1,2,3]*, Maria S. Khrabrova[1,4], Elena S. Vashukova[5], Andrey S. Glotov[5], Anastasia O. Anpilova[1,6], Vladimir A. Dobronravov[6], Maria E. Velizhanina[1], Aleksandr A. Rubel[1,7]*

1 Laboratory of Amyloid Biology, Saint Petersburg University, St. Petersburg, Russia, 2 Laboratory of Toxinology and Molecular Systematics, L.A. Orbeli Institute of Physiology, National Academy of Sciences of the Republic of Armenia, Yerevan, Armenia, 3 Laboratory of Comparative Behavioral Genetics, Pavlov Institute of Physiology, Russian Academy of Sciences, St. Petersburg, Russia, 4 Department of Propaedeutics of Internal disease, Pavlov University, St. Petersburg, Russia, 5 Department of Genomic Medicine, D.O. Ott Research Institute of Obstetrics, Gynecology and Reproductology, St. Petersburg, Russia, 6 Research Institute of Nephrology, Pavlov University, St. Petersburg, Russia, 7 Department of Genetics and Biotechnology, St. Petersburg State University, St. Petersburg, Russia

* serg900@yandex.ru (SAF); a.rubel@spbu.ru (AAR)

**Data Availability Statement:** All relevant data are within the manuscript and its Supporting information files.

## Abstract

Preeclampsia is a disorder that can occur during pregnancy and is one of the leading causes of death among pregnant women. This disorder occurs after the 20th week of pregnancy and is characterized by arterial hypertension, proteinuria, fetoplacental, and multiple organ dysfunctions. Despite the long history of studying preeclampsia, its etiology and pathogenesis remain poorly understood, and therapy is symptomatic. One of the factors of the disorder is believed to be misfolded proteins that are prone to form amyloid aggregates. The CRD tests, utilizing the binding of the amyloid-specific dye Congo red to urine components, demonstrate high efficiency in diagnosing preeclampsia. However, these tests have also been found to be positive in other disorders with proteinuria, presumably associated with concomitant amyloidosis. To assess the limitations of the CRD tests, we examined urine congophilia and protein components mediating Congo red positivity in patients with proteinuria, including preeclampsia, amyloid and non-amyloid nephropathies. We stained the urine samples and calculated congophilia levels. We also assessed the contribution of large protein aggregates to congophilia values using ultracentrifugation and determined the molecular weights of congophilic urinary proteins using centrifugal concentrators. All proteinuric groups demonstrate positive results in the CRD tests and congophilia levels were more than two times higher compared with the control non-proteinuric groups ($p < 0.01$). There was a strong correlation between urine protein excretion and congophilia in amyloid nephropathy ($r_s = 0.76$), non-amyloid nephropathies ($r_s = 0.90$), and preeclampsia ($r_s = 0.90$). Removal of large aggregates from urine did not affect the congophilia levels. Separation of urine protein fractions revealed congophilic components in the range of 30–100 kDa, including monomeric serum albumin. Our results indicate limitations of CRD tests in preeclampsia diagnostics in women with renal disorders and

**Funding:** This study was supported by grant 22-25-00315 from Russian Science Foundation (MEV). https://rscf.ru/project/22-25-00315/ The funders had no role in study design, data collection and analysis, decision to publish, or preparation of the manuscript.

**Competing interests:** The authors have declared that no competing interests exist.

underscore the need for further research on the mechanisms of Congo red binding with urine components.

## Introduction

Preeclampsia is a severe complication of pregnancy that is characterized by elevated blood pressure, proteinuria, seizures, failure of multiple organs, and even death [1–7]. An accurate and early diagnosis of this condition is an unmet clinical need [8, 9]. In 2014, Buhimschi et al. demonstrated that urine proteins in patients with preeclampsia exhibited properties similar to amyloid [10]. For diagnostic purposes, they developed the Congo red dot (CRD) test, which assessed the retention of the amyloid-specific Congo red dye by urine proteins on a membrane (Congo red retention, CRR). Samples from preeclampsia patients retained the dye, while those from non-preeclampsia patients lost it when washed off (Fig 1A). For use in clinical practice modified versions of the CRD test have been introduced known as CRD paper test [11, 12] where positive test results are determined by the area of the spot (Fig 1B). When tested in pregnant women, this modified test provided a sensitivity of 80.2% and specificity of 89.2%, making it more effective predictor than well-known disorder markers such as fms-like tyrosine kinase 1 and placental growth factor [11].

McCarthy et al. assessed the effectiveness of CRD tests in pregnant and non-pregnant women with chronic kidney disease (CKD) with unspecified morphology, hypertension, and lupus nephritis [13]. The authors confirmed the effectiveness of the test in diagnosing preeclampsia and noted its ease of use and low cost. However, this study found a strong correlation between congophilia and protein:creatinine ratio (Spearman rank correlations, 0.702). The authors also showed that the median CRR did not differ in the groups of pregnant women with preeclampsia and pregnant women with CKD. Both groups were characterized by high values of proteinuria. Moreover, the CRR in the group of non-pregnant women with lupus nephritis was higher than that in non-pregnant women with systemic lupus erythematosus without nephritis. The authors concluded that there may be limitations in using the CRD test in patients with various etiologies of proteinuria. Rood et al. suggested that false positive CRD test results in patients with proteinuria could be associated with the development of renal amyloidosis, a pre-Alzheimer's state or preeclampsia in subclinical state [11]. Whether the renal

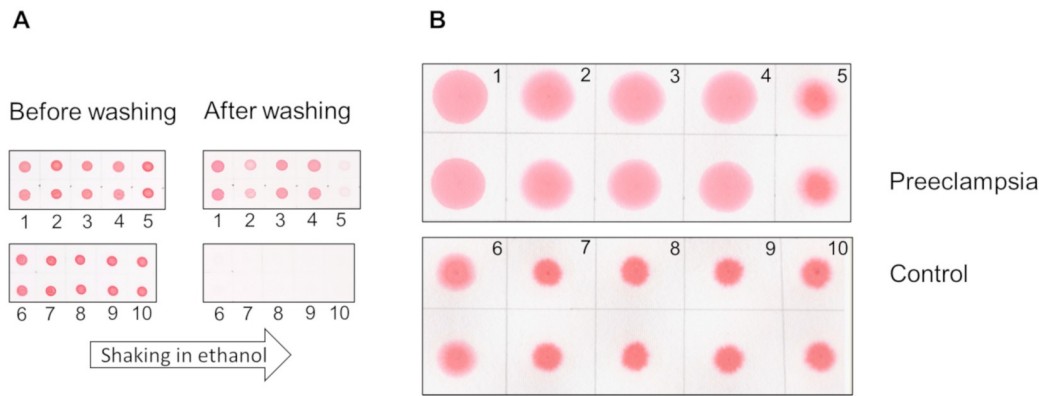

**Fig 1. The representation of the CRD tests' results.** The CRD membrane (A) and CRD paper (B) tests are shown. The representative images displayed are from five pregnant women with preeclampsia (1–5) and five control pregnant women (6–10).

amyloidoses and non-amyloid proteinuric nephropathies yield different CRD test results remains unknown. Another poorly investigated aspect [10] is whether urinary congophilia in patients with preeclampsia originates from the urinary proteins themselves or from amyloid-like aggregates. In order to address these gaps and identify potential limitations of the CRD test, we investigated congophilia and protein components that mediate Congo red positivity in urine samples from patients with proteinuria of different etiology, including preeclampsia and histologically confirmed amyloid and non-amyloid nephropathies.

## Materials and methods

### Study design, patient selection, and data collection

An institution-based case-control study was conducted. Strengthening the Reporting of Observational Studies in Epidemiology (STROBE) was used while reporting the study. Two study groups comprised patients with amyloid nephropathies (AN) and non-amyloid nephropathies (NA), confirmed by detailed clinical and histological evaluation. All patients with morphologically confirmed renal amyloidosis and proteinuria >0.3 g/l who were admitted to nephrology clinic of Pavlov University during the sample collection period were included in the study. For every AN patient enrolled we immediately recruited the patient with non-amyloid nephropathy who matched by the age and proteinuria value in order to avoid any selection bias. The diagnostic performance of CRD tests varies greatly and these differences are largely associated with different approaches to the diagnosing preeclampsia. If preeclampsia is diagnosed in the absence of proteinuria, the sensitivity and specificity of the test are lower [14]. Therefore, to evaluate the specificity of the CRD test, it is important to analyze study groups (AN vs NA) with similar level of proteinuria.

Routine renal morphology analysis in all AN and NA included light microscopy (staining for hematoxylin-eosin, Jones silver stain, Masson, Congo-rot, Periodic acid-Schiff), immunofluorescence on frozen sections for immunoglobulin (Ig) G, IgA, IgM, C3, fibrinogen, light chain (LC) kappa, LC lambda. Immunohistochemistry with pronase digestion was performed if needed as well as immunoperoxidase staining on paraffin-embedded tissue sections for serum amyloid A, lambda and kappa LC, prealbumin for amyloid typing. Electron microscopy was additionally carried out for renal pathology clarification. All histopathological samples were examined by experienced renal pathologist. Patients with a great clinical suspicion on the amyloidosis with renal involvement were excluded if the kidney biopsy was not done due to contraindications. The sample size was 27 and determined by assuming a confidence level of 99%, a margin of error of 5% and population proportion 1% (the proportion of amyloidoses in the population of nephropathies with proteinuria).

We also enrolled women with preeclampsia (PE) as the positive controls for CRD tests. Preeclampsia was diagnosed according to the clinical guidelines [7]. Inclusion criteria for PE were systolic blood pressure >140 mm and/or diastolic blood pressure >90 mm after the 20th week of pregnancy in combination with proteinuria. Women with normal pregnancy (NP) and patients with various non-proteinuric kidney diseases (Control) served as negative controls for CRD tests as used in previous studies [11, 15]. Inclusion criteria for Control and NP were 24-h albuminuria <30 mg and estimated glomerular filtration rate (eGFR) >60 ml/min/1,73m$^2$, which confirms the absence of a hidden renal pathology. At the time of enrollment, pregnancy was excluded in all women from the AN, NA, and Control groups. Serum creatinine was used to evaluate kidney function in pregnant women [16], while the eGFR was additionally calculated using the CKD-EPI 2009 formula [17] for the AN, NA, and Control groups.

All participants signed the informed consent as a part of Regular Medical documentation. The study was approved either by the Ethics Committees of Pavlov University (St. Petersburg,

Russia) (No.: 21–250; date of approval 28 June 2021) or the Research Institute of Obstetrics (St. Petersburg, Russia) (No.: 97; date of approval 27 June 2019). Urine samples for this study were collected from 14.09.2019 to 30.12.2021 for PE and NP groups and from 12.09.2021 to 28.12.2021 for AN, NA, and Control groups.

### Congophilia assays

The urine samples were stored at -80˚C, thawed on ice, and centrifuged at 4000 x$g$, 4˚C for 5 min before the experiments. Before testing, all samples were assessed for protein levels using the Quick Start Bradford Protein Assay kit (Bio-Rad, USA). Congophilia tests were performed as previously described [10, 11] with some modifications. 80 μl of each sample was mixed with 8 μl of a 0.2% aqueous solution of Congo red dye (Merck, Germany), incubated for 10 min and applied on a nitrocellulose membrane (Amersham Protran 0.45 NC, GE Healthcare, USA) in two aliquots of 2 μl (membrane test) and FN 3 chromatography paper (thickness: 0.19 mm; square weight: 90 g/m$^2$) in two aliquots of 40 μl (paper test). The membrane was dried for 10 minutes, then wetted with water. It was imaged using a camera in a lab-made box. Next, it was incubated for 1 min in 50% and 70% ethanol, and washed for 1 h in 90% ethanol with shaking. After rinsing successively in 70% ethanol, 50% ethanol, and water, the membrane was imaged again. The integrated intensity of each spot was determined using ImageJ (version 1.51j8) and the CRR was calculated as the ratio of the average intensities after washing and before washing. In the paper test, the stained samples were scanned 15 min after application on the paper, and the area of the spots was calculated in ImageJ as Congo red area (CRA, px).

For congophilia tests and polyacrylamide gel electrophoresis (PAGE), the following proteins were used: human serum albumin (HSA) (A3782, Sigma-Aldrich, USA), bovine serum albumin (BSA) (23209, Thermo Fisher Scientific, USA), and bovine gamma globulin (BGG) (500–0208, Bio-Rad, USA).

### Separation of urine fractions

The separation of high molecular weight (MW) aggregates was carried out using an Optima MAX-XP ultracentrifuge (Beckman Coulter, USA) at 300,000 x$g$ and 4˚C for 2 h [18]. To isolate protein fractions within specific MW ranges, 200 μl of urine samples were applied onto Amicon Ultra-4 Centrifugal Filter Devices (Merck Millipore, Germany) with a cut-off of 30 kDa (UFC8030) and 100 kDa (UFC8100), followed by centrifugation at 7,000 x$g$ for 10 min. Subsequently, 190 μl of filtrates and 10 μl of concentrates were collected (concentration factor 20). Concentrates were diluted by phosphate buffer, pH 7.4 (S-P4417, Merck, Germany) to 200 μl, concentrated once more (washing), and diluted to the volume of the original sample.

### Urine protein concentrating

Four control urine samples from pregnant women with protein levels less than 0.3 mg/ml were concentrated by lyophilization and dissolving the sediment in small volumes of distilled water, followed by centrifugation of the resulting suspension for 10 min at 4000 x$g$. The volume of the supernatant was approximately equal to that of the precipitate, and its protein concentration was approximately 2.5 times lower than the concentration in the samples before lyophilization multiplied by the concentration factor. Thus, the majority of the protein after dissolution remained in the sediment. The supernatant was analyzed in membrane CRD test and 10% PAGE followed by Coomassie brilliant blue staining. Spectra Multicolor Broad Range Protein Ladder (26634, Thermo Scientific, USA) was used to assess MW of proteins separated in PAGE.

## Statistics

Data are presented as median with interquartile range or mean ± standard deviation for continuous variables and frequencies with % for categorical variables. Parameters among the groups were compared by analysis of variance. For continuous and categorical variables comparison, the Mann-Whitney U-test and the chi-square test were applied, respectively. The mean CRRs, CRAs, and proteinuria were compared between groups by a two-sided randomization test in the Drosophila Courtship Lite v. 1.3 [19, 20]. The 95% confidence intervals (C. I.) for means were calculated by bootstrapping (10,000 iterations) [21]. The 95% C. I. and $p$-values for the Spearman rank-order correlation coefficient were assessed by $t$-test. To determine the linear approximation (coefficient of determination, $R^2$) Microsoft Excel 2016 was used. Statistical significance was assumed at $p < 0.01$.

## Results

### Patient description

AN group (n = 27) presented with 5 cases of serum amyloid A amyloidosis and 22 cases of immunoglobulin light chain (AL)-amyloidosis. NA group (n = 27) included patients with focal segmental glomerulosclerosis (n = 5), non-amyloid type of monoclonal immunoglobulin-related kidney disease (n = 5), membranous nephropathy (n = 4), diabetic nephropathy (n = 4), immunoglobulin A nephropathy (n = 4), anti-neutrophil cytoplasmic antibodies associated glomerulonephritis (n = 2), lupus nephritis (n = 1), C3-glomerulopathy (n = 1), and idiopathic membranoproliferative glomerulonephritis (n = 1). The control group comprised patients with diabetes mellitus (n = 13), cardiovascular disease (n = 12), systemic autoimmune disorder (n = 2), aplastic anemia (n = 2), Cushing disease (n = 1), human immunodeficiency virus (n = 1), multiple myeloma (n = 1), and four healthy persons. Demographic and clinical findings of the studied groups are presented in Table 1. The groups did not differ significantly in descriptive parameters. Since patients in the AN group were on average 10 years older than in the AN group ($p > 0.01$), we assessed the correlation between CRR values and age to avoid potential biases in congophilia estimates in these groups. We did not find significant Spearman correlation values in both groups and therefore the influence of patient age on congophilia is excluded in our data.

**Table 1. Demographic and clinical data in studied groups.**

| | AN (n = 27) | NA (n = 27) | Control (n = 36) | PE (n = 13) | NP (n = 31) |
|---|---|---|---|---|---|
| **Descriptive parameters**[a] | | | | | |
| **Woman/man, n/n** | 18/9 | 15/12 | 24/12 | 13/0 | 31/0 |
| **Age, years** | 59±11 | 49±15 | 48±17 | 32±4 | 32±4 |
| **BMI, kg/m²** | 24.2 (20.4; 25.9) | 25.4 (22.1; 30.3) | 27.2 (23.0; 31.0) | 30 (26; 36) | 25 (24; 28) |
| **Renal findings**[b] | | | | | |
| **Serum creatinine, mg/dl** | 1.13 (0.9; 2.77) | 1.55 (0.9; 2.23) | **0.81 (0.69; 0.94)** | 0.76 (0.7; 0.8) | 0.73 (0.68; 0.81) |
| **eGFR CKD-EPI, ml/min/1,73m²** | 54 (23; 86) | 43 (27; 95) | **85 (76; 102)** | n/a | n/a |
| **24-h proteinuria, g/24-h** | 8 (5.6;12.4) | 6.6 (3.6; 13.1) | n/a | 5 (2.5; 5.5) | n/a |
| **24-h Albuminuria, mg/24-h** | n/a | n/a | 8 (6;14) | n/a | 10 (6; 15) |

AN, amyloid nephropathies; BMI, body mass index; Control, patients without proteinuria; eGFR, estimated glomerular filtration rate; NA, non-amyloid nephropathies; NP, normal pregnancy; PE, preeclampsia.

[a] No parameters differed significantly between AN and NA when compared in pairs, as well as between PE and NP.

[b] In the ANOVA analysis, the Control group significantly differed from AN and NA in terms of eGFR and serum creatinine (in bold, $p < 0.001$).

## Levels of congophilia and proteinuria in the study groups

All proteinuric groups demonstrate positivity in the CRD tests. The mean CRAs in the AN and NA groups were 2.5 and 2.2 times higher, respectively, compared with the Control group (Fig 2A). Similarly, the mean CRA was two-fold higher in PE compared with the NP group. However, the mean CRAs did not differ between AN and NA groups. This trend was consistent when comparing mean CRR in the membrane test (Fig 2B) and protein concentrations between groups (Fig 2C).

The urine protein excretion and CRR values strongly correlated in AN, NA, and PE groups (Fig 3).

To examine Congo red binding to the main urine proteins, we conducted the CRD membrane tests on a series of dilutions of HSA and BGG in phosphate buffer, pH 7.4, and revealed a positive correlation between CRR and the concentration of these proteins (Fig 4A and 4B). The average ratio of CRRs to protein levels in HSA samples and the PE group did not differ (Fig 4C).

## Congophilia of different urine fractions

The contribution of large urine protein aggregates to CRR value and the MW of the urine proteins binding to Congo red were estimated. After isolating the aggregates from urine samples by ultracentrifugation, the CRR values of the supernatants still were high. The mean CRRs did not differ between supernatants after centrifugation at 300,000 x$g$ and 4,000 x$g$ ($p$ = 0.432, Fig 5). Thus, the results of the experiment did not reveal a significant contribution of large aggregates to the congophilia values of the analyzed samples.

Separation of protein fractions in HSA solution and urine samples using the centrifugal concentrators revealed congophilic components within the 30–100 kDa range (Fig 6A). According to PAGE analysis, the main two protein bands in the urine samples were around 45–50 and 70 kDa (Fig 6B).

## Congophilia levels and protein compositions in concentrated urine

The effectiveness of the CRD tests was previously assessed by utilizing concentrated urine samples from healthy pregnant women as control subjects [10]. We compared the CRR values

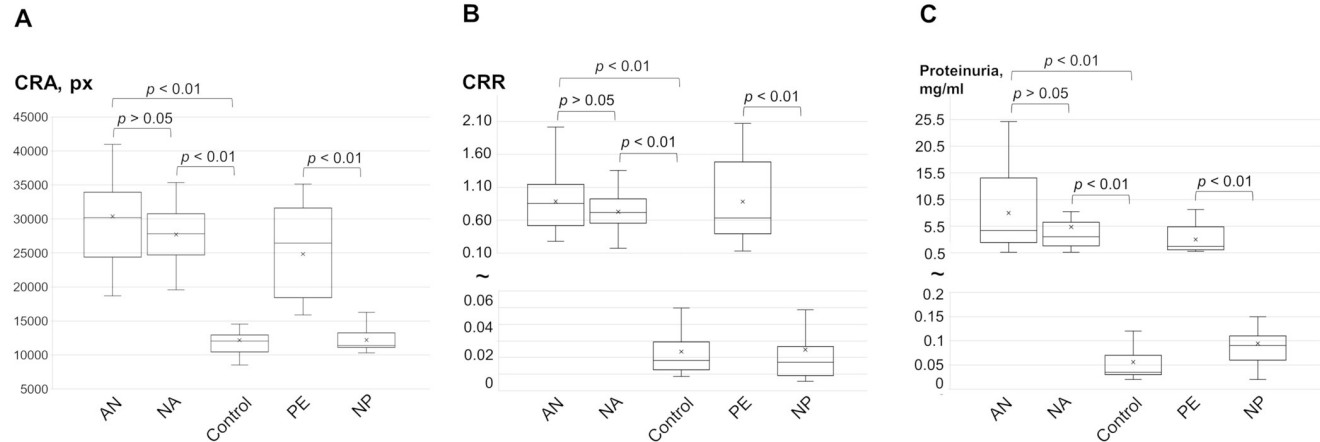

**Fig 2. The results of the CRD tests and proteinuria levels in studied groups.** The values of congophilia observed in the paper (A) and membrane (B) tests, as well as the levels of proteinuria (C), are shown. The box plots present the 25th and 75th percentiles (box), the maximum and minimum values, the median (line in the box), and the mean (cross). AN, amyloid nephropathies; Control, patients without proteinuria; CRA, Congo red area; CRR, Congo red retention; NA, non-amyloid nephropathies; NP, normal pregnancy; PE, preeclampsia.

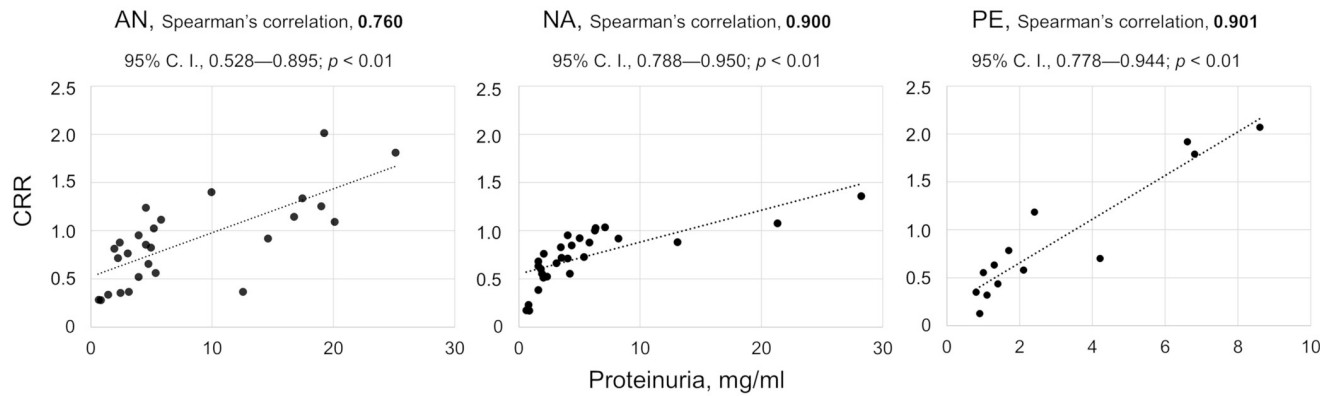

**Fig 3. The quantitative relationship between urine congophilia and proteinuria in three groups with nephropathies.** The Spearman rank-order correlation coefficients (Spearman's correlation), 95% confidence intervals (C. I.), $p$ values ($t$-test), and scatter charts with linear trend lines are shown for each group. AN, amyloid nephropathies; CRR, Congo red retention; NA, non-amyloid nephropathies; PE, preeclampsia.

between concentrated urine samples and an HSA solution with the same protein concentration (Fig 7A) given that the ratio of congophilia to protein is equal in non-concentrated samples and the HSA solutions (Fig 3). The membrane test showed a lower CRR in each concentrated sample compared to an equal concentration of HSA (Fig 7B). PAGE analysis of the original and concentrated samples revealed no apparent qualitative changes in the protein composition. In nearly all cases, a distinct band, presumably attributed to HSA, was observed (Fig 7C). Thus, concentrated urine samples had lower CRRs than HSA samples, but there were no obvious changes in the protein composition of the samples after concentration.

## Discussion

Tests for congophilia in diagnosing preeclampsia are simple, cheap, and fast for primary screening of pregnant women [14, 22]. Although there are no reported limitations in the use of

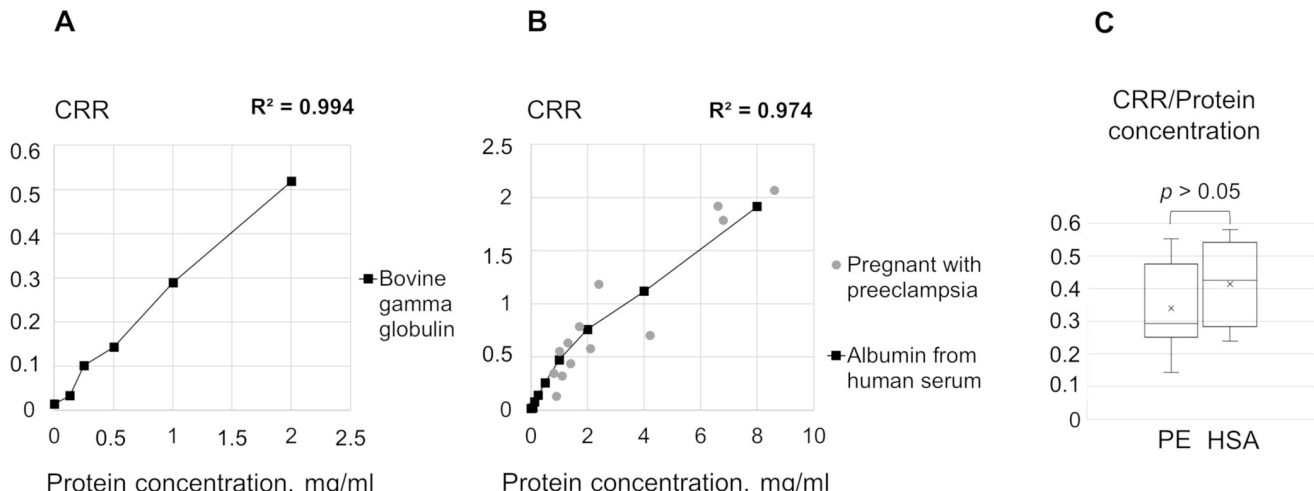

**Fig 4. The quantitative relationship between congophilia and concentrations of BGG and HSA in comparison with preeclampsia samples.** The reliability of the linear approximation (coefficient of determination, $R^2$) of CRR dependence on concentrations of BGG (A) and HSA (B) is shown. Section B also shows urinary congophilia in the PE group (grey circles). (C) The ratios of the CRRs to the protein concentrations in HSA solutions and PE samples are shown. CRR, Congo red retention; BGG, bovine gamma globulin; HSA, human serum albumin; PE, preeclampsia.

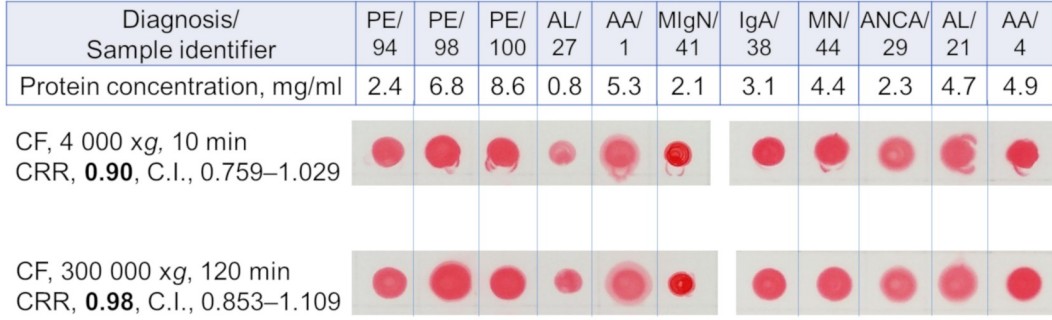

**Fig 5. Congophilia of urine samples from patients of different groups before and after ultracentrifugation.** On the left, sample preparation conditions, mean CRRs (n = 11), and 95% C. I. are indicated. AA, serum amyloid A amyloidosis; AL, Immunoglobulin light chain amyloidosis; ANCA, anti-neutrophil cytoplasmic antibody-associated glomerulonephritis; CF, centrifugation; CRR, Congo red retention; IgA, Immunoglobulin A nephropathy; MIgN, monoclonal immunoglobulin-related kidney disease; MN, membranous nephropathy; PE, preeclampsia.

the tests, the effectiveness of these methods may vary widely among hospitals in different countries [14]. In our study, we the first evaluated urine congophilia using the CRD test in morphologically verified proteinuric nephropathies, including non-amyloid diseases and amyloidosis. The latter is assumed to be more prone to urine congophilia which is stated to depend not on the concentration of protein in the urine but is associated with the processes of amyloidogenesis [11]. We found that the CRD test yielded positive results in all proteinuric patients, regardless of the etiology of proteinuria (Fig 2). Urinary congophilia was strongly correlated with the urine protein concentration (Fig 3). The positivity of the CRD test in cases with proteinuria limits its clinical utility for diagnosing preeclampsia. Thus, pregnant women with pre-existing renal disease unrelated to pregnancy can yield positive results in the CRD tests. Conversely, when proteinuria is absent in cases of preeclampsia, the CRD test may also have reduced effectiveness, as evidenced by a recent meta-analysis [23].

Previous studies have suggested a formation of amyloid-like protein aggregates in urine as substrates for Congo red binding [10]. We have also shown that aggregates resistant to

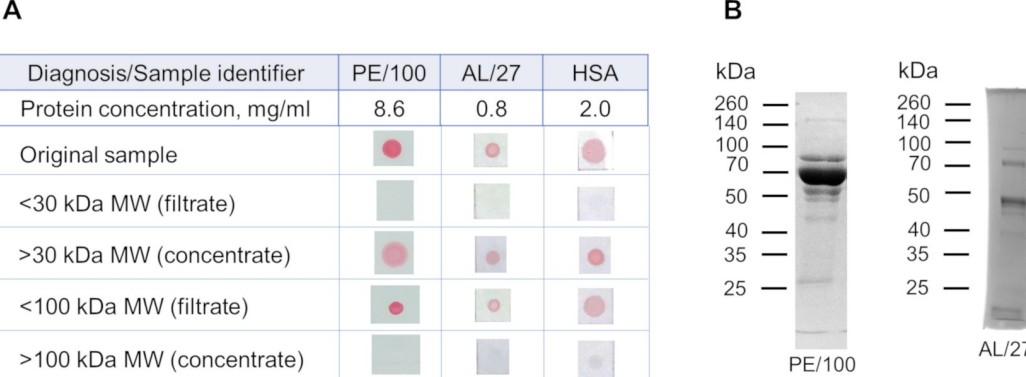

**Fig 6. Congophilia of urine samples and HSA solution after centrifugation on concentrators with cut-offs of 30 and 100 kDa.** (A) Samples before centrifugation (original samples) and after centrifugation (concentrates and filtrates) are analyzed using a membrane test. At the left, urine protein fractions are listed. (B) 10% polyacrylamide gel electrophoresis of urine samples with 15 μg of protein is shown. Proteins in the gel are stained by Coomassie brilliant blue. AL, Immunoglobulin light chain amyloidosis; HSA, human serum albumin; MW, molecular weight; PE, preeclampsia.

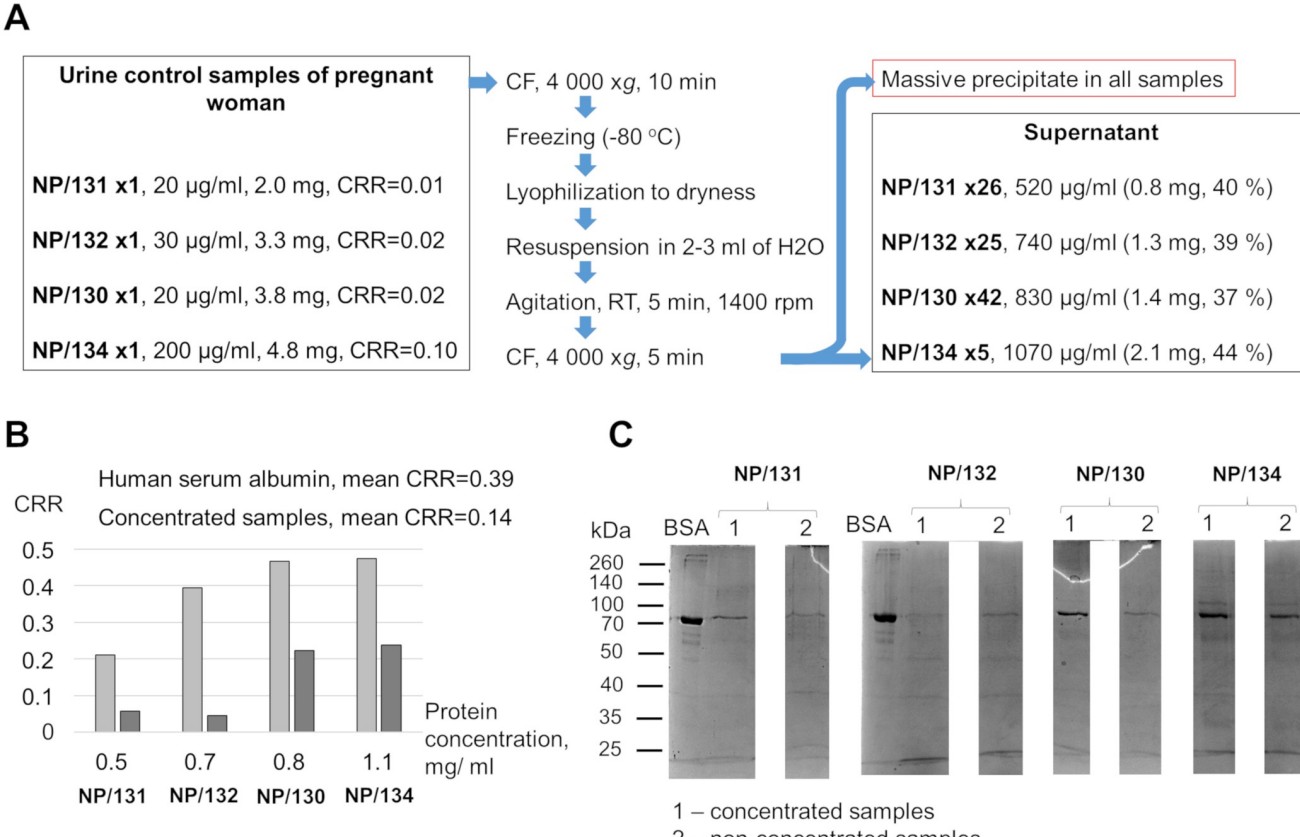

**Fig 7. Congophilia and protein composition in concentrated urine samples from women with uncomplicated pregnancy.** (A) From left to right: initial protein concentrations, protein amounts and CRRs for the four control urine samples (NP/130-134); procedure scheme; protein concentrations and total amounts of protein (weights and percentages of the initial amount) in the supernatant obtained by concentrating. (B) Mean CRRs of concentrated samples and HSA solutions at concentrations equal to concentrated samples are shown. (C) 10% polyacrylamide gel electrophoresis results for original and concentrated samples (1 μg of protein) are shown. BSA (2 μg) was applied to localize the putative HSA in the analyzed samples. Proteins in the gel are stained by Coomassie brilliant blue. BSA, bovine serum albumin; CF, centrifugation; CRR, Congo red retention; NP, control sample from a woman with uncomplicated pregnancy; RT, room temperature.

detergent treatment, possibly with amyloid structure, can be precipitated from urine samples with proteinuria by ultracentrifugation [24]. We have attempted to estimate the size of such aggregates that bind Congo red. In our experiments, urinary congophilia was associated with urine proteins with a molecular weight less than 100 kDa (Fig 6), and the CRD test was still positive after removing possible aggregates of urine proteins (Fig 5). These findings suggest that large protein aggregates are unlikely to be responsible for the CRD test positivity (Fig 4) and are typical in urines in various nephropathies [25, 26], these proteins might be responsible for the urinary congophilia in any proteinuric conditions. The binding of Congo red to HSA monomers [27] and the presence of urinary congophilia only in the 30–100 kDa fraction (Fig 6A) support this assumption. However, we cannot entirely rule out the presence of low-molecular protein aggregation either under specific urine microenvironments or induced by conditions of the CRD test.

To demonstrate the efficacy of the CRD test previous studies compared urinary congophilia in samples from preeclampsia patients with samples from normally pregnant women after equalizing the protein levels in the samples from both groups [10, 13, 15]. We found that the

congophilia values of concentrated urine from healthy pregnant women were lower in comparison with a solution of monomeric HSA (Fig 7). This difference is not associated with a change in the protein composition after concentration, despite the precipitation of a significant proportion of proteins. We speculate that concentrated samples may have a reduced proportion of misfolded proteins, which are more prone to precipitation. It has been shown that denatured albumin has a greater affinity for Congo red [28]. Therefore, concentrated urine with a lower proportion of misfolded proteins will show lower levels of congophilia compared to an unconcentrated urine sample at the same concentration. Thus, using concentrated urine as control samples may lead to inaccurate assessment of the congophilia test performance.

The strength of this exploratory case-control study lies in its combination of experimental and clinical approaches. Unlike other studies, the detailed clinical and histological examination provides precise renal diagnoses, enabling the inclusion wide spectrum of amyloid and non-amyloid nephropathy cases in the evaluation of the clinical significance of the CRD test. This approach, along with easily reproducible experiments, convincingly demonstrates the non-specificity of the CRD test in patients with proteinuria. However, we could not include pregnant women with chronic kidney disease and proteinuric disease unrelated to pregnancy (i.e., glomerulonephritis) that require separate studies. Additionally, we did not employ mass spectrometry and other molecular techniques for urine proteome analysis, which could have provided valuable insights into possible mechanisms of urine congophilia. This is an area that could be explored in future research to further enhance our understanding of the CRD test and its clinical implications.

## Conclusion

Our study showed that urinary congophilia used to diagnose preeclampsia is largely determined by protein concentration and is not associated with the presence of large protein aggregates in the sample. Positive test results were obtained in patients with renal disorders of various etiologies, regardless of the presence of amyloid deposits in the body. Our findings indicate limitations of CRD tests in diagnosing preeclampsia in women with renal disorders or in cases without proteinuria and shed light on the possible reasons for the inaccurate estimation of the CRD test effectiveness. Further research focusing on molecular mechanisms of Congo red binding to specific urine proteins could propose the possible clinical applications of the congophilia phenomenon in preeclampsia and other nephropathies.

## Supporting information

**S1 Appendix. Experimental data.** All data used to generate the results for this study. (XLSX)

**S1 Raw images. All images used to generate the figures for this study.** (PDF)

## Acknowledgments

The authors wish to express their gratitude to Dr. Yury O. Chernoff from the School of Biological Sciences at Georgia Institute of Technology, USA, for his invaluable consultation, constructive criticism, and insightful comments. Additionally, the authors extend their appreciation to Anton Radaev from the Research Resource Center "Chromas" of St. Petersburg State University, and Elizaveta J. Gorodilova from the Center for Molecular and Cell Technologies of St. Petersburg State University, for their valuable technical support throughout the study.

## Author Contributions

**Conceptualization:** Sergei A. Fedotov, Andrey S. Glotov, Aleksandr A. Rubel.

**Data curation:** Sergei A. Fedotov, Maria S. Khrabrova, Aleksandr A. Rubel.

**Formal analysis:** Sergei A. Fedotov, Maria S. Khrabrova, Elena S. Vashukova.

**Funding acquisition:** Maria E. Velizhanina, Aleksandr A. Rubel.

**Investigation:** Sergei A. Fedotov, Elena S. Vashukova, Anastasia O. Anpilova.

**Project administration:** Sergei A. Fedotov, Maria S. Khrabrova, Andrey S. Glotov, Maria E. Velizhanina, Aleksandr A. Rubel.

**Resources:** Elena S. Vashukova, Andrey S. Glotov, Aleksandr A. Rubel.

**Supervision:** Sergei A. Fedotov, Aleksandr A. Rubel.

**Validation:** Sergei A. Fedotov, Elena S. Vashukova.

**Visualization:** Sergei A. Fedotov, Maria S. Khrabrova.

**Writing – original draft:** Sergei A. Fedotov, Maria S. Khrabrova.

**Writing – review & editing:** Sergei A. Fedotov, Maria S. Khrabrova, Vladimir A. Dobronravov, Maria E. Velizhanina, Aleksandr A. Rubel.

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
