## [Decision Letter · Decision Letter 0]

30 Nov 2023

PONE-D-23-29876Diagnostics of preeclampsia based on Congo red binding to urinary components: rationales and limitationsPLOS ONE

Dear Dr. Fedotov,

Thank you for submitting your manuscript to PLOS ONE. After careful consideration, we feel that it has merit but does not fully meet PLOS ONE’s publication criteria as it currently stands. Therefore, we invite you to submit a revised version of the manuscript that addresses the points raised during the review process.

We look forward to receiving your revised manuscript.

Kind regards,

Giovanni Tossetta, Ph.D

Academic Editor

PLOS ONE

Journal Requirements:

Reviewers' comments:

Reviewer's Responses to Questions

**Comments to the Author**

1. Is the manuscript technically sound, and do the data support the conclusions?

Reviewer #1: Yes

Reviewer #2: Partly

Reviewer #3: Yes

2. Has the statistical analysis been performed appropriately and rigorously? 

Reviewer #1: Yes

Reviewer #2: No

Reviewer #3: Yes

3. Have the authors made all data underlying the findings in their manuscript fully available?

Reviewer #1: Yes

Reviewer #2: Yes

Reviewer #3: Yes

4. Is the manuscript presented in an intelligible fashion and written in standard English?

Reviewer #1: Yes

Reviewer #2: No

Reviewer #3: No

5. Review Comments to the Author

**Reviewer #1**: i have gone through the manuscript and noted that it is technically sound and well written. It attempted to add to the avalanche of simple and cheap diagnostic tests for early detection of preeclampsia which unfortunately remain a disease of many theories. I recommend its acceptance for publication in PLOS ONE.

**Reviewer #2**: The need for a sensitive tool to predict, detect, and manage pre-eclampsia makes this work crucial to the literature. The method section needs additional attention. Please note the comments on this section. The authors should follow STROBE guidelines. The following are my opinions and concerns:

The abstract section: the authors need to note the following within this section:

• Line #35-39: Please describe the results regarding the groups (experimental and control) and summarise their relative prevalence of proteinuria or Congo red test positivity.

• Please include the method and material section in the abstract.

The Introduction section:

• Line #59-67 Clarify the study's rationale and objective, making it more relevant to diagnosing pre-eclampsia.

The methods section: To enhance the study’s reproducibility, please note the following suggestions:

• Please speculate on the study design. Setting (its relevance for this study). Clarify the study population and speculate on the sample size determination.

• Line #70-71: How were the participants included in the study? Which of them were excluded? How were the experimental and control groups selected? Were any precautions taken during the selection of participants?

• How were the participants included in the study? Which of them were excluded?

The Results section: How many participants were involved or analysed in the study?

• Lines# 142, 167 and 187: please rephrase the subheading into a more concise and precise one.

• Line# 169-172, please summarise the actual results. The narrations on the results should be done in the discussion section.

• Line# 188-200: please summarise the actual results. Consider a section of this part within experimental methods.

The Discussion section:

Lines #211-212: were all the subjects nephropathic?

Lines #240-242: Were there any limitations?

The Conclusion:

Lines #240-242: Kindly provide your conclusions. These results come from different research.

Tables and Figures

• Figure 2 A-C: The title and abbreviations used in each figure must be explained below the outputs.

• Table 1: It is difficult to understand the table right away. How similar or different are the groups according to the clinical and demographic data? Both summary and inferential statistical indices are in the same table. Can they be in separate tables?

**Reviewer #3**: Overall, this is a clearly written manuscript. The introduction is relevant and based on previous studies. Regarding the basic flow of ideas overall all subsections follow a logical order and sufficient information about the previous study findings is presented for readers to follow the present study rationale and procedures. However, the research question and objective of the research is phrased differently in different sections, though the reader can understand that the objective was to show that congophilia test can be positive in other kidney diseases as well and therefore cannot be relied on as the most sensitive predictor of preeclampsia.

The methods are generally appropriate, although clarification of a few details and provision of a rationale for the use of this test. Overall, the results are clear and compelling with two possible exceptions. In the control group for the patients with amyloidosis and non-amyloidosis renal diseases, most of the patients chosen had different conditions and 4 patients had no diseases. Can the results of this control group be correctly interpreted compared to the non-control group? Can this be replicated? Also, the age groups of patients with amyloidosis are not like the other control and non-control groups which may influence the results.

The authors do make an efficient contribution to the research literature in this area of investigation. The nature of the work is original, and they have reemphasized what has been already present in the literature adding renal amyloidosis patients to the study. Summarizes the overall picture and synthesizes the knowledge gained adequately.

Overall, this is an agreeable and good manuscript that has propositions emphasizing the limitations of using congophilia as a sole screening or diagnostic test for preeclampsia. Well written with appropriate style and language. Citation and paraphrasing are used appropriately except for one wrong citation Few errors but they do not impede readers' understanding remarkably.

6. PLOS authors have the option to publish the peer review history of their article (what does this mean?). If published, this will include your full peer review and any attached files.

Reviewer #1: No

Reviewer #2: No

Reviewer #3: No

---

## [Author Response · Author response to Decision Letter 0]

27 Dec 2023

Dear Dr. Giovanni Tossetta,

We are grateful to reviewers for critical reading of the manuscript and their comments. Accordingly, we have revised the manuscript essentially. The changes in the manuscript are highlighted. Responses to reviewers' comments are submitted. Changes were made to Figure 5, 6 and 7 to clarify sample identifiers. All corrections did not change the results of the experiments depicted in the figures.

We ensured that our manuscript meets PLOS ONE's style requirements, including those for file naming (according to The PLOS ONE style templates https://journals.plos.org/plosone/s/file?id=wjVg/PLOSOne_formatting_sample_main_body.pdf)

Upon re-submitting our revised manuscript, we uploaded our study’s minimal underlying data set as Supporting Information file “S1_Appendix.xlsx”and the original images underlying all gel/blot results as Supporting Information file “S2_raw_images.pdf”.

Sincerely,

Dr. Sergei Fedotov

Responses to comments of Reviewer #1

General comments and suggestions: i have gone through the manuscript and noted that it is technically sound and well written. It attempted to add to the avalanche of simple and cheap diagnostic tests for early detection of preeclampsia which unfortunately remain a disease of many theories. I recommend its acceptance for publication in PLOS ONE.

Answer: Thank you for taking the time to review our manuscript. We appreciate your positive feedback and recognition of the technical soundness and quality of writing. We also share your concern about the lack of simple and affordable diagnostic tests for early detection of preeclampsia. We hope that our research will contribute to the ongoing efforts in understanding and combating preeclampsia.

 

Responses to comments of Reviewer #2

Thank you for your careful review of our manuscript and for providing constructive feedback. We appreciate your suggestions for improvement, and we have made the revisions in response to your comments and suggestions.

General comments and suggestions: The need for a sensitive tool to predict, detect, and manage pre-eclampsia makes this work crucial to the literature. The method section needs additional attention. Please note the comments on this section. The authors should follow STROBE guidelines. 

Answer: We agree that the need for a sensitive tool to predict, detect, and manage pre-eclampsia is crucial, and we appreciate your recognition of the importance of our work. Regarding the method section, we have carefully reviewed your comments and have made additional revisions to address your concerns. Specifically, we have paid close attention to following the STROBE guidelines to ensure the clarity and transparency of our methodology.

The following are my opinions and concerns:

(1) Questions / requests: The abstract section: the authors need to note the following within this section. Line #35-39: Please describe the results regarding the groups (experimental and control) and summarise their relative prevalence of proteinuria or Congo red test positivity. Please include the method and material section in the abstract.

Answer: Thank you for your feedback. A description of the results regarding the groups and method used has been added to the abstract (lines #34-42).

(2) Questions / requests: The Introduction section: Line #59-67 Clarify the study's rationale and objective, making it more relevant to diagnosing pre-eclampsia.

Answer: We have made the necessary clarifications in the Introduction section of the paper to better explain the study's rationale and objective, specifically addressing the relevance to diagnosing pre-eclampsia. Please refer to lines #63-74 for the updated information.

(3) Questions / requests: The methods section: To enhance the study’s reproducibility, please note the following suggestions. Please speculate on the study design. Setting (its relevance for this study). Clarify the study population and speculate on the sample size determination. Line #70-71: How were the participants included in the study? Which of them were excluded? How were the experimental and control groups selected? Were any precautions taken during the selection of participants? How were the participants included in the study? Which of them were excluded?

Answer: Thank you for your suggestions. We have made the necessary additions to the Methods section of the paper to address your concerns and enhance the study's reproducibility. Specifically, we have provided information on the study design and setting (lines #85-93), clarified the study population and sample size determination (lines #94-104), and described the selection of participants for the experimental and control groups, as well as any precautions taken during their selection (lines #85-87, 100-102, 106-111). Please refer to these lines for the updated information.

(4) Questions / requests: The Results section: How many participants were involved or analysed in the study? Lines# 142, 167 and 187: please rephrase the subheading into a more concise and precise one. Line # 169-172, please summarise the actual results. The narrations on the results should be done in the discussion section. Line# 188-200: please summarise the actual results. Consider a section of this part within experimental methods.

Answer: We have made the necessary revisions to address your concerns. We have provided the number of participants involved in the study in the results section (lines #169-170), rephrased the subheadings to be more concise and precise (lines #190, 215 and 236), and summarized the actual results in each subsection (lines #191, 219-221 and 244-246). Additionally, we have included a description of the experimental methods used to obtain the results in the subsection "Congophilia level and protein composition in concentrated urine" (line #236) in the Methods section (lines #147-153).

(5) Questions / requests: The Discussion section.

Lines #211-212: were all the subjects nephropathic?

Answer: Yes, all the subjects in our study had nephropathy, as mentioned in lines #259-261 of the manuscript.

Lines #240-242: Were there any limitations?

Answer: We have added information about any limitations in the study. This can be found in lines #258-259 and #299-302 of the revised manuscript.

(5) Questions / requests: The Conclusion: Lines #240-242: Kindly provide your conclusions. These results come from different research.

Answer: The requested corrections have been made to the Conclusion section (lines #308-311).

(6) Questions / requests: Tables and Figures.

Figure 2 A-C: The title and abbreviations used in each figure must be explained below the outputs.

Answer: Dear reviewer, sorry, but the title and abbreviations were explained in the figure legend.

Table 1: It is difficult to understand the table right away. How similar or different are the groups according to the clinical and demographic data? Both summary and inferential statistical indices are in the same table. Can they be in separate tables?

Answer: Thank you for bringing these points to our attention. Summary and inferential statistical indices are separated. Significant differences are now highlighted in bold to make it easier to understand the table at a glance.

 

Responses to comments of Reviewer #3

We greatly appreciate your thorough and insightful review of our manuscript. We have taken each of your points into careful consideration and have made the necessary revisions to address all of the concerns raised.

General comments and suggestions: Overall, this is a clearly written manuscript. The introduction is relevant and based on previous studies. Regarding the basic flow of ideas overall all subsections follow a logical order and sufficient information about the previous study findings is presented for readers to follow the present study rationale and procedures. However, the research question and objective of the research is phrased differently in different sections, though the reader can understand that the objective was to show that congophilia test can be positive in other kidney diseases as well and therefore cannot be relied on as the most sensitive predictor of preeclampsia.

Answer: We have made improvements to better define the research question and purpose in our paper (lines #32-34 and #63-74).

(1) Questions / requests: The methods are generally appropriate, although clarification of a few details and provision of a rationale for the use of this test.

Answer: Thank you for bringing this to our attention. We have now provided a clarification of the criteria for preeclampsia in lines #106-108 and have included references for congophilia assays in lines #109 and #123-124. These additions will help provide a rationale for the use of this test and ensure the clarity of our methods. 

(2) Questions / requests: Overall, the results are clear and compelling with two possible exceptions. In the control group for the patients with amyloidosis and non-amyloidosis renal diseases, most of the patients chosen had different conditions and 4 patients had no diseases. Can the results of this control group be correctly interpreted compared to the non-control group? Can this be replicated? 

Answer: We have added references to previous studies where negative CRD test results in patients without proteinuria were demonstrated (lines #109). These non-proteinuric patients serve as methodological controls to determine the lower limit of CRD test positivity values. In our study, we specifically analyze data in proteinuric groups, so the possible effect of heterogeneity in the control group should not influence the conclusions drawn in our work.

(3) Questions / requests: Also, the age groups of patients with amyloidosis are not like the other control and non-control groups which may influence the results.

Answer: Thank you for your feedback. We have taken into consideration the age inconsistency issue and have made the necessary amendments in the manuscript (lines #178-182).

(4) Questions / requests: The authors do make an efficient contribution to the research literature in this area of investigation. The nature of the work is original, and they have reemphasized what has been already present in the literature adding renal amyloidosis patients to the study. Summarizes the overall picture and synthesizes the knowledge gained adequately.

Overall, this is an agreeable and good manuscript that has propositions emphasizing the limitations of using congophilia as a sole screening or diagnostic test for preeclampsia. Well written with appropriate style and language. Citation and paraphrasing are used appropriately except for one wrong citation Few errors but they do not impede readers' understanding remarkably.

Answer: Thank you for your thorough review and positive assessment of our article. We greatly appreciate your attention to detail and valuable suggestions. After careful consideration, we have incorporated all of your recommendations and made the necessary corrections throughout the manuscript (highlighted, lines #284-293). We have replaced the incorrect citation with the correct one (lines #375-377) and hope that we have corrected any other errors that were present in the paper. Your feedback has been instrumental in improving the quality and accuracy of our work.

---

## [Editor Report · Decision Letter 1]

29 Dec 2023

Diagnostics of preeclampsia based on Congo red binding to urinary components: rationales and limitations

PONE-D-23-29876R1

Dear Dr. Fedotov,

We’re pleased to inform you that your manuscript has been judged scientifically suitable for publication and will be formally accepted for publication once it meets all outstanding technical requirements.

Kind regards,

Giovanni Tossetta, Ph.D

Academic Editor

PLOS ONE

---

## [Editor Report · Acceptance letter]

10 Jan 2024

PONE-D-23-29876R1 

PLOS ONE

Dear Dr. Fedotov, 

I'm pleased to inform you that your manuscript has been deemed suitable for publication in PLOS ONE. Congratulations! Your manuscript is now being handed over to our production team.

Kind regards, 

on behalf of

Dr. Giovanni Tossetta 

Academic Editor

PLOS ONE